# Associations between smoking to relieve stress, motivation to stop and quit attempts across the social spectrum: A population survey in England

Olga Perski◯*, Maria Theodoraki, Sharon Cox, Loren Kock, Lion Shahab◯, Jamie Brown◯

Research Department of Behavioural Science and Health, University College London, London, United Kingdom

* olga.perski@ucl.ac.uk

## Abstract

Smoking prevalence in several high-income countries is steadily declining but remains persistently high in 'lower' socioeconomic position (SEP) groups, contributing to inequities in morbidity and mortality. Smoking to relieve stress is a commonly endorsed motive for continued smoking; however, it remains unclear whether smoking to relieve stress has a negative impact on motivation to stop and future quit attempts and if so, whether associations are moderated by SEP. This was an observational study with cross-sectional and prospective survey data from the nationally representative Smoking Toolkit Study in England. A total of 1,135 adult smokers were surveyed at baseline, with 153 (13.5%) respondents followed up at 12 months. Respondents provided information on demographic, social and smoking characteristics. A series of multivariable logistic regression analyses was conducted. Bayes Factors (BFs) were calculated to explore non-significant associations. Smoking to relieve stress was commonly endorsed by respondents from both 'lower' (43.2% [95% CI = 39.4%, 47.0%]) and 'higher' (40.5% [95% CI = 35.9%, 45.1%]) SEP groups ($p = 0.39$). Smoking to relieve stress was associated with high motivation to stop at baseline ($OR_{adj} = 1.48$, 95% CI = 1.03–2.12, $p = 0.035$) but not significantly with the odds of making a quit attempt at a 12-month follow-up, although the magnitude and direction of the effect was similar to that observed for high motivation to stop ($OR_{adj} = 1.49$, 95% CI = 0.69–3.20, $p = 0.3$). Data were insensitive to detect moderation effects of SEP (BF = 0.90 and BF = 1.65, respectively). Smoking to relieve stress is a commonly endorsed motive and is associated with high motivation to stop but not significantly with the odds of making a quit attempt in the next 12 months, although the magnitude and direction of the effect was similar for both outcomes. There was no clear evidence of moderation by SEP, although data were insensitive to distinguish the alternative from the null hypothesis.

**Data Availability Statement:** The dataset underpinning the analyses is available via Figshare: https://figshare.com/articles/dataset/Associations_

between_smoking_to_relieve_stress_motivation_
to_stop_and_future_quit_attempts_across_the_
social_spectrum_A_population_survey_in_
England/14731029.

**Funding:** Cancer Research UK is the main
contributor (C1417/A22962), but the UK
Department of Health, Pfizer, GlaxoSmithKline and
Johnson and Johnson have also all contributed
funding to the data collection for the Smoking
Toolkit Study. The Society for the Study of
Addiction and the National Institute for Health
Research's School for Public Health Research have
contributed funding to the data collection for the
Alcohol Toolkit Study. JB, LS, OP, SC and LK are
members of SPECTRUM, a UK Prevention
Research Partnership Consortium (MR/S037519/
1). UKPRP is an initiative funded by the UK
Research and Innovation Councils, the Department
of Health and Social Care (England) and the UK
devolved administrations, and leading health
research charities. The funders had no final role in
the study design; in the collection, analysis and
interpretation of data; in the writing of the report; or
in the decision to submit the paper for publication.
All researchers listed as authors are independent
from the funders and all final decisions about the
research were taken by the investigators and were
unrestricted.

**Competing interests:** OP, MT, SC and LK have no
conflicts of interest to declare. LS has received a
research grant and honoraria for a talk and travel
expenses from manufacturers of smoking
cessation medications (Pfizer and Johnson &
Johnson). JB has received unrestricted research
funding from Pfizer to study smoking cessation.
This does not alter our adherence to PLOS ONE's
policies on sharing data and materials.

## Introduction

Tobacco smoking remains one of the leading causes of premature illness and death, with global smoking-related deaths approximating 8 million people per year in 2019 [1]. Smoking prevalence in a number of high-income countries, including England, has been steadily declining for the past decade but remains persistently high in 'lower' socioeconomic position (SEP) groups [2]. In England, 'lower' SEP is associated with increased morbidity and mortality, of which smoking is a significant contributor [3]. To help reduce observed health inequities between 'higher' and 'lower' SEP groups [4], supporting people to stop smoking–particularly those from priority SEP groups–is urgently needed. The development of effective policies and interventions requires the identification of key barriers to smoking cessation, such as beliefs or motives that hinder smoking cessation attempts.

Motivational factors, including desire or intention to stop, have been identified as key predictors of smoking cessation attempts [5]. Although longitudinal studies show that smoking cessation (as compared with continued smoking) is associated with reduced stress, anxiety and depression–with equal improvements to mental wellbeing observed in adults with and without a mental health diagnosis [6]–approximately 40% of smokers in England indicate that they smoke to cope with stress or anxiety [7]. However, it remains unclear whether smoking to relieve stress undermines motivation to stop and subsequent quit attempts. According to the PRIME (Plans, Responses, Impulses, Motives, Evaluations) Theory of Motivation [8], plans to stop smoking influence behaviour only if they generate sufficiently strong impulses at relevant moments in time. However, competing motives (including the belief that smoking helps to relieve stress) may act to undermine both plans and subsequent quit attempts by generating strong, opposing impulses. It is therefore plausible that the belief that smoking relieves stress may lead to reduced motivation to stop or it may undermine intentions to stop by generating strong, competing impulses which contribute to continued smoking. Previous research has examined whether the belief that one is addicted to smoking ('perceived addiction') undermines motivation to stop and future quit attempts. However, contrary to predictions, perceived addiction to smoking was recently found to be positively associated with motivation to stop [9] and future quit attempts [10].

SEP refers to social and economic factors that influence the position that individuals inhabit within the structures of society, and which in turn influence their health [11]. These factors typically include (but are not limited to) income, educational attainment, home ownership/ tenure, occupation and employment status. The term 'lower' SEP is used in this study to reflect the relatively average lower income and status of those in routine and manual occupational trades and those who are unemployed. Smokers from lower SEP groups face multiple environmental and social stressors, including financial and occupational hardship and instability, as well as poor physical and mental health [11]. They also face smoking-specific stressors such as a lack of social support and higher nicotine dependence (compared with smokers from more advantaged groups) [12, 13], which means that negative affect from withdrawal is likely experienced as more intense and stressful, also highlighting the role of negative affect control through continued smoking [14].

Several qualitative studies have highlighted the role of smoking to cope with stress in particular subgroups of smokers, including adults experiencing homelessness [15], smokers living with psychiatric conditions [16], heavy smokers [17] and older smokers [18]. However, it remains unclear whether smoking to relieve stress is more commonly endorsed in smokers from lower (compared with higher) SEP groups, but it is plausible given the frequent occurrence of high stress levels in lower SEP groups, coupled with the widely held belief that smoking relieves stress. In addition, the belief that smoking is beneficial for stress is likely weighed

against perceived benefits of quitting, such as improved health and financial savings (which could help alleviate financial hardship) [19]. It is therefore important to examine not only whether smoking to relieve stress has a negative impact on motivation to stop and subsequent attempts to quit, but also whether these associations are moderated by SEP. Recent research indicates that although motivation to stop appears similar by housing tenure–a reliable indicator of SEP and predictor of smoking status [20]–and prevalence of past-year quit attempts are higher in adults living in homes that belong to a housing association or are rented from local authority (compared with other housing), people from lower SEP groups are less likely to succeed in quitting [21].

We therefore aimed to examine, in the nationally representative Smoking Toolkit Study, the associations of smoking to relieve stress with motivation to stop (measured at baseline) and quit attempts at a 12-month follow-up in England, and whether these associations are moderated by SEP. Specifically, we aimed to address the following research questions:

1. Do smokers who indicate that they smoke to relieve stress have reduced odds of high (vs. low) motivation to stop smoking, without and with adjustment for age, sex, SEP, children in the household, cigarettes smoked per day and number of quit attempts in the past year?

2. Do smokers who indicate that they smoke to relieve stress have reduced odds of making a quit attempt at a 12-month follow-up, without and with adjustment for age, sex, SEP, children in the household, cigarettes smoked per day and number of quit attempts in the past year?

3. Does SEP moderate any associations between smoking to relieve stress and a) high motivation to stop and b) quit attempts at a 12-month follow-up, without and with adjustment for age, sex, SEP, children in the household, cigarettes per day and number of quit attempts in the past year?

## Methods

### Study design and setting

This was an observational study involving cross-sectional and prospective survey data. The STROBE guidelines were used in the design and reporting of this study [22].

The study is part of the ongoing Smoking Toolkit Study (STS), which involves monthly, face-to-face, computer-assisted household surveys of adults aged 16+ in England [23, 24]. The sample is a hybrid of a random probability and quota sample, which results in a sample that is representative of the adult population of smokers in England. Interviews are held with one household member in selected geographic output areas until quotas are fulfilled. The quotas are based on factors influencing the probability of being at home (i.e. working status, age and gender). This hybrid form of random probability and quota sampling is considered superior to conventional quota sampling. Here, the choice of households to approach is limited by the random allocation of small output areas and rather than being sent to specific households in advance, interviewers can choose which households within these small geographic areas are most likely to fulfil their quotas. Therefore, unlike random probability sampling, it is not appropriate to record the response rate in the STS. Data included in the present study were collected across four waves (139–142) from respondents surveyed between April and July 2018 (when smoking to relieve stress was included in the STS). Attempts to recontact consenting past-year smokers via telephone are made 12 months later. Follow-up respondents are not remunerated. Up to eight attempts are made to follow up each consenting respondent, with calls made at different times of day across weekdays and weekends.

Initial analyses were conducted as part of a student project. An extended study protocol and analysis plan were pre-registered on the Open Science Framework prior to conducting the analyses (https://osf.io/efbn2).

## Inclusion criteria

Respondents were included in the analyses if they were aged 16+ years and smoked cigarettes at the time of the baseline survey. Respondents who indicated that they did not smoke cigarettes but did smoke tobacco of some kind were not included (~0.4%) as they do not complete equivalent measures of dependence (e.g. cigarettes per day). Respondents with missing data on any of the variables of interest were excluded from the analyses.

## Ethics statement

Ethical approval was granted by UCL's Research Ethics Committee (0498/001). Respondents provided verbal informed consent prior to each interview.

## Measures

**Outcome variables.** The first outcome variable was motivation to stop, measured at baseline with the validated Motivation To Stop Scale [25, 26]. Respondents are asked: "Which of the following best describes you?" Response options include: 1) I don't want to stop smoking; 2) I think I should stop smoking but don't really want to; 3) I want to stop smoking but haven't thought about when; 4) I REALLY want to stop smoking, but I don't know when I will; 5) I want to stop smoking and hope to soon; 6) I REALLY want to stop smoking and intend to in the next 3 months; 7) I REALLY want to stop smoking and intend to in the next month. To aid interpretation, responses were dichotomised into low (response options 1–5) and high (response options 6–7) motivation to stop, as previously done in studies using this variable [27, 28].

The second outcome variable was quit attempts at a 12-month follow-up, measured with a single item: "How many serious attempts to stop smoking have you made in the last 12 months? By serious attempt I mean you decided that you would try to make sure you never smoked again." Responses were coded 0 for respondents who reported no quit attempt and 1 for those who reported 1+ quit attempts.

**Explanatory variable.** The explanatory variable was smoking to relieve stress, measured at baseline with a single item asking respondents to select from a list the motives that they believe are important in keeping them smoking. Those endorsing the response option "It helps me cope with stress or anxiety" were coded as 1, the rest as 0.

**Covariates.** Covariates measured at baseline included: age (16–24 years, 25–34 years, 35–44 years, 45–54 years, 55–64 years, 65+ years); sex (male, female); SEP, operationalised as occupational social grade and measured with the British National Readership Survey's Social Grade Classification Tool [29], which comprises the categories AB (higher and intermediate managerial, administrative and professional), C1 (supervisory, clerical and junior managerial, administrative and professional), C2 (skilled manual workers), D (semi-skilled and unskilled manual workers) and E (state pensioners, casual and lowest-grade workers, unemployed with state benefits), with responses grouped into 'lower' (C2DE) and 'higher' (ABC1) social grade; children in the household (no vs. yes); cigarettes per day (converted to daily consumption for non-daily smokers who reported the number of cigarettes smoked per week); and number of quit attempts in the past year (0, 1, 2, 3, 4+).

We also captured housing tenure as an alternative indicator of SEP [20]. Respondents whose homes belonged to a housing association or were rented from local authority were

categorised as 'lower' SEP. All other responses (i.e. houses which were 'bought on a mortgage', 'owned outright by household', 'rented from private landlord', 'other') were categorised as 'higher' SEP.

### Data analysis

**Analyses were conducted in RStudio v.1.2.5033.** Data were weighted with the rim (marginal) technique [30] to match the analytic sample to the proportions of the English population profile on the dimensions of age, social grade, region, housing tenure, ethnicity and working status within sex. We report descriptive statistics for the baseline and follow-up samples. Any differences between the baseline and follow-up samples were assessed with Chi-square or *t*-tests, as appropriate.

To address RQ1 and RQ2, unweighted logistic regression analyses were conducted, without and with adjustment for all covariates. To address RQ3, the analyses were repeated, including the two-way interaction between smoking to relieve stress and SEP. As a planned sensitivity analysis, the analyses were repeated with housing tenure used as the indicator of SEP [20].

**Bayes Factors.** In the event of non-significant interactions, we had planned to calculate Bayes Factors (using an online calculator; www.bayesfactor.info) to examine whether the non-significant associations could best be characterised as evidence of no association, evidence of an association, or whether the data were insensitive to detect an association. As there was limited evidence to inform what effect sizes to expect for the potential interactions with SEP, we set the expected effect sizes to OR = 0.7, informed by previous research examining the association between enjoyment of smoking and future quit attempts [31]. The alternative hypothesis was conservatively represented by a half-normal distribution. Bayes Factors (BFs) can be interpreted as follows: BFs >3 lend support to the alternative hypothesis (over the null), BFs <1/3 lend support to the null, and BFs in-between reflect data being insensitive to distinguish the two.

## Results

A total of 1,203 cigarette smokers were surveyed between April and July 2018, of whom 1,135 (94.3%; unweighted) had complete data on all variables of interest. A total of 153 (13.5%; unweighted) respondents provided data at the 12-month follow-up (see Table 1). Compared with the baseline sample, those responding to the 12-month follow-up survey were older ($p < 0.001$), more likely to be from higher SEP groups ($p < 0.001$) and less likely to have children in the household ($p = 0.02$). At baseline, a total of 43.2% (95% CI = 39.4%, 47.0%) of respondents from lower SEP groups indicated that they smoke to relieve stress, compared with 40.5% (95% CI = 35.9%, 45.1%) of respondents from higher SEP groups ($p = 0.39$).

In unadjusted and adjusted analyses without the interaction, smoking to relieve stress was associated with 48% greater odds of high motivation to stop ($OR_{adj} = 1.48$, 95% CI = 1.03–2.12, $p = 0.035$; see Table 2). The association between smoking to relieve stress and quit attempts at the 12-month follow-up was not significant ($OR_{adj} = 1.49$, 95% CI = 0.69–3.20, $p = 0.3$). The adjusted associations between the exposure and outcome variables were unchanged in the sensitivity analyses with housing tenure (see S1 Table). In an unplanned sensitivity analysis with cigarettes per day coded as a categorical variable, the associations were similar (see S2 Table).

In unadjusted and adjusted analyses, the two-way interactive effects of smoking to relieve stress and SEP on high motivation to stop ($OR_{adj} = 1.66$, 95% CI = 0.14–24.3, $p = 0.7$) and quit attempts at the 12-month follow-up ($OR_{adj} = 0.31$, 95% CI = 0.1–1.4, $p = 0.12$) were not significant (see S3 and S4 Tables for the ORs stratified by SEP). Results remained non-significant in

**Table 1. Demographic, social and smoking characteristics of current cigarette smokers at baseline and at the 12-month follow-up.**

| | Baseline sample | | Follow-up sample | | |
| --- | --- | --- | --- | --- | --- |
| | Unweighted (N = 1,135) | Weighted (N = 1,170) | Unweighted (N = 153) | Weighted (N = 152) | *p*-value** |
| **Sex, N (%)** | | | | | 0.78 |
| Women | 557 (49%) | 550 (47%) | 73 (48%) | 69 (45%) | |
| Men | 578 (51%) | 620 (53%) | 80 (52%) | 84 (55%) | |
| **Age, N (%)** | | | | | **<0.001** |
| 16–24 years | 189 (17%) | 194 (17%) | 13 (8.5%) | 13 (8.2%) | |
| 25–34 years | 242 (21%) | 287 (25%) | 20 (13%) | 24 (16%) | |
| 35–44 years | 191 (17%) | 224 (19%) | 27 (18%) | 34 (22%) | |
| 45–54 years | 204 (18%) | 206 (18%) | 30 (20%) | 29 (19%) | |
| 55–64 years | 152 (13%) | 133 (11%) | 27 (18%) | 24 (15%) | |
| 65+ years | 157 (14%) | 125 (11%) | 36 (24%) | 29 (19%) | |
| **SEP*, N (%)** | | | | | **<0.001** |
| Higher | 457 (40%) | 455 (39%) | 84 (55%) | 82 (54%) | |
| Lower | 678 (60%) | 715 (61%) | 69 (45%) | 71 (46%) | |
| **Housing tenure, N (%)** | | | | | 0.44 |
| Higher | 767 (68%) | 905 (77%) | 108 (71%) | 121 (79%) | |
| Lower | 368 (32%) | 264 (23%) | 45 (29%) | 31 (21%) | |
| **Children in the household, N (%)** | | | | | **0.02** |
| No | 760 (67%) | 774 (66%) | 116 (76%) | 114 (75%) | |
| Yes | 375 (33%) | 396 (34%) | 37 (24%) | 38 (25%) | |
| **Cigarettes per day, M (SD)** | 10.6 (8.0) | 10.3 (7.8) | 11.4 (8.1) | 10.8 (7.7) | 0.25 |
| **Number of quit attempts in the past year, N (%)** | | | | | 0.17 |
| 0 | 827 (73%) | 859 (73%) | 119 (78%) | 119 (78%) | |
| 1 | 182 (16%) | 184 (16%) | 23 (15%) | 21 (14%) | |
| 2 | 68 (6.0%) | 74 (6.3%) | 9 (5.9%) | 10 (6.6%) | |
| 3 | 33 (2.9%) | 31 (2.6%) | 0 (0%) | 0 (0%) | |
| 4+ | 25 (2.2%) | 22 (1.9%) | 2 (1.3%) | 2 (1.1%) | |
| **Smoking to relieve stress, N (%)** | | | | | 0.58 |
| No | 657 (58%) | 686 (59%) | 85 (56%) | 87 (57%) | |
| Yes | 478 (42%) | 484 (41%) | 68 (44%) | 65 (43%) | |
| **Motivation to stop smoking, N (%)** | | | | | 0.32 |
| Low | 968 (85%) | 1,004 (86%) | 135 (88%) | 136 (89%) | |
| High | 167 (15%) | 166 (14%) | 18 (12%) | 17 (11%) | |
| **Quit attempt at the 12-month follow-up, N (%)** | | | | | |
| No | - | - | 89 (58%) | 88 (58%) | - |
| Yes | - | - | 64 (42%) | 64 (42%) | |

* SEP = socioeconomic position

** *p*-values pertain to comparisons of the baseline and follow-up samples (unweighted), with values <0.05 highlighted in bold

the planned sensitivity analyses with housing tenure ($OR_{adj}$ = 0.34, 95% CI = 0.03–4.31, *p* = 0.4 and $OR_{adj}$ = 0.72, 95% CI = 0.14–3.8, *p* = 0.7, respectively).

## Bayes Factors (BFs)

The calculation of BFs indicated that the data on the two-way interactive effects of smoking to relieve stress and SEP on high motivation to stop and quit attempts at the 12-month follow-up were insensitive to detect an association (BF = 0.90 and BF = 1.65, respectively). As an

**Table 2. Unadjusted and adjusted odds ratios for the associations of smoking to relieve stress with i) high motivation to stop and ii) quit attempts at the 12-month follow-up.**

| | High motivation to stop (N = 1,135) | | | Quit attempts at the 12-month follow-up (N = 153) | | |
|---|---|---|---|---|---|---|
| | OR^ | 95% CI | *p*-value | OR | 95% CI | *p*-value |
| **Smoking to relieve stress (ref = No)** | | | | | | |
| Yes | 1.75 | 1.26, 2.43 | **<0.001** | 1.64 | 0.86, 3.16 | 0.13 |
| **Sex (ref = Women)** | | | | | | |
| Men | 0.80 | 0.57, 1.11 | 0.2 | 0.31 | 0.09, 0.87 | **0.033** |
| **Age (ref = 16–24 years)** | | | | | | |
| 25–34 years | 1.16 | 0.71, 1.94 | 0.6 | 3.00 | 0.38, 62.8 | 0.4 |
| 35–44 years | 1.06 | 0.62, 1.83 | 0.8 | 2.09 | 0.27, 43.3 | 0.5 |
| 45–54 years | 0.84 | 0.49, 1.47 | 0.5 | 1.33 | 0.15, 28.5 | 0.8 |
| 55–64 years | 0.64 | 0.33, 1.20 | 0.2 | 1.50 | 0.17, 32.1 | 0.7 |
| 65+ years | 0.42 | 0.20, 0.83 | **0.016** | 1.09 | 0.13, 23.2 | >0.9 |
| **SEP* (ref = Higher)** | | | | | | |
| Lower | 0.69 | 0.50, 0.97 | **0.030** | 0.57 | 0.19, 1.56 | 0.3 |
| **Children in the household (ref = No)** | | | | | | |
| Yes | 1.39 | 0.99, 1.95 | 0.055 | 0.16 | 0.01, 0.83 | 0.082 |
| **CPD** ** | 0.96 | 0.93, 0.98 | **<0.001** | 0.93 | 0.86, 1.00 | 0.074 |
| **Number of quit attempts in the past year (ref = 0)** | | | | | | |
| 1 | 5.72 | 3.83, 8.55 | **<0.001** | 5.35 | 1.70, 16.5 | **0.003** |
| 2 | 6.50 | 3.68, 11.3 | **<0.001** | 3.49 | 0.47, 17.3 | 0.2 |
| 3 | 5.96 | 2.68, 12.6 | **<0.001** | - | - | - |
| 4+ | 6.71 | 2.74, 15.5 | **<0.001** | - | - | - |
| | **High motivation to stop (N = 1,135)** | | | **Future quit attempt (N = 153)** | | |
| | OR_adj | 95% CI | p-value | OR_adj | 95% CI | p-value |
| **Smoking to relieve stress (ref = No)** | | | | | | |
| Yes | 1.48 | 1.03, 2.12 | **0.035** | 1.49 | 0.69, 3.20 | 0.3 |
| **Sex (ref = Women)** | | | | | | |
| Men | 0.96 | 0.67, 1.38 | 0.8 | 1.32 | 0.62, 2.85 | 0.5 |
| **Age (ref = 16–24 years)** | | | | | | |
| 25–34 years | 1.04 | 0.60, 1.82 | 0.9 | 0.61 | 0.12, 2.97 | 0.5 |
| 35–44 years | 0.97 | 0.53, 1.77 | >0.9 | 0.40 | 0.08, 1.80 | 0.2 |
| 45–54 years | 1.01 | 0.55, 1.83 | >0.9 | 0.50 | 0.11, 2.19 | 0.4 |
| 55–64 years | 0.96 | 0.47, 1.89 | 0.9 | 0.21 | 0.04, 0.98 | 0.051 |
| 65+ years | 0.54 | 0.25, 1.14 | 0.11 | 0.33 | 0.07, 1.42 | 0.14 |
| **SEP* (ref = Higher)** | | | | | | |
| Lower | 0.72 | 0.50, 1.04 | 0.084 | 1.30 | 0.63, 2.70 | 0.5 |
| **Children in the household (ref = No)** | | | | | | |
| Yes | 1.18 | 0.78, 1.78 | 0.4 | 0.93 | 0.37, 2.30 | 0.9 |
| **CPD** ** | 0.96 | 0.94, 0.99 | **0.007** | 1.00 | 0.96, 1.05 | 0.8 |
| **Number of quit attempts in the past year (ref = 0)** | | | | | | |
| 1 | 5.29 | 3.50, 8.00 | **<0.001** | 6.09 | 2.20, 19.0 | **<0.001** |
| 2 | 6.08 | 3.40, 10.7 | **<0.001** | 5.63 | 1.20, 40.6 | **0.044** |
| 3 | 5.41 | 2.37, 11.8 | **<0.001** | - | - | - |
| 4+ | 7.13 | 2.83, 17.1 | **<0.001** | 3.23 | 0.11, 90.9 | 0.4 |

* SEP = socioeconomic position

** CPD = cigarettes per day

^ = Odds Ratio

OR_adj = adjusted for sex, age, SEP, children in the household, CPD and number of quit attempts in the past year

- could not be estimated due to low cell counts

unplanned sensitivity analysis, we also calculated a BF for the non-significant association between smoking to relieve stress and quit attempts at the 12-month follow-up using the same prior as the above analyses, which indicated that data were insensitive to detect an association (BF = 0.46).

## Discussion

### Principal findings

This study aimed to examine whether smoking to relieve stress is associated with high motivation to stop at baseline and quit attempts at a 12-month follow-up, and whether any associations are moderated by SEP. Contrary to expectations, we found that smoking to relieve stress was associated with high motivation to stop smoking, but not significantly with the odds of making a quit attempt at the 12-month follow-up. Although non-significant, the direction and magnitude of the association of smoking to relieve stress with future quit attempts were similar to those observed for the association with high motivation to stop. We did not observe significant moderation by SEP; however, the Bayes Factors indicated that data were insensitive to detect an association (as opposed to lending support to the null hypothesis of no association). For this to be known, other studies with larger samples need to be conducted to detect any potential moderation effects by SEP.

The finding that smoking to relieve stress is associated with high (as opposed to low) motivation to stop is consistent with previous research indicating that the belief that one is addicted to smoking is positively associated with motivation to stop [9]. There are several plausible explanations for this finding. First, people who smoke to relieve stress may evaluate their smoking experience as biological or instrumental (as opposed to pleasurable), which may be perceived as unpleasant or annoying, hence leading to greater motivation to stop. Second, those who smoke to relieve stress may find smoking enjoyable but at the same time, find it a source of dissonance to use smoking as a 'crutch' to cope with stress, hence leading to greater motivation to stop. Third, it is also possible that there is a common factor (e.g., depression) that gives rise both to the belief that smoking helps to relieve stress and high motivation to stop. Previous research has found that smokers with more severe (compared with less severe) levels of depression report lower quitting self-efficacy (which may be influenced by positive outcome expectations, such as the belief that smoking helps to cope with stress) but higher motivation to stop [32]. Fourth, as both smoking to relieve stress and motivation to stop were measured at the same time, there is potential for reverse causation, whereby those highly motivated to stop are also more likely to identify motives for smoking that some people may judge negatively (e.g. they may judge it unpleasant to feel that smoking is needed to relieve stress).

The finding that smokers from both higher and lower SEP groups endorse the belief that smoking helps to relieve stress may indicate that smokers are not only more/less likely to endorse this belief depending on their level of actual stressors (which may, on average, be higher in smokers from lower SEP groups). Instead, this may be related to people's perceived ability to cope with stressors. Smokers across the social spectrum may *perceive* themselves to be better/worse equipped to cope with stress (irrespective of whether they actually are), thus influencing their beliefs about the role of smoking (e.g. positive outcome expectancies).

### Strengths and limitations

This study was strengthened by examining associations in a nationally representative sample of smokers in England. To our knowledge, this was the first study to examine the associations of smoking to relieve stress, motivation to stop and future quit attempts in addition to differences in smoking to relieve stress by SEP. However, our study also had several limitations.

First, the sample size for the prospective association of smoking to relieve stress and future quit attempts was small, which led to imprecision. However, this was pre-determined by the period for which the funding was available (i.e., the question about smoking motives–including smoking to relieve stress–was included only in a few STS waves), and we attempted to mitigate this by calculating Bayes Factors to elucidate whether the data provided evidence of no association or whether they were insensitive to detect an association. Second, the follow-up sample differed from the baseline sample on important characteristics, including SEP. This means that, had our findings pertaining to the follow-up sample been statistically significant, they may not be generalisable to the general population of smokers. In addition, if those who were more likely to be followed up (i.e. older smokers, men, those from higher SEP groups) were also more or less likely to endorse the belief that they smoke to relieve stress, this may have distorted any association with the outcome. Third, although the Smoking Toolkit Study is a nationally representative survey on key sociodemographic and smoking characteristics, it may not be representative with regards to psychological characteristics such as motivation to stop or the motive that smoking helps to relieve stress and it also may not reach smokers who are the most disadvantaged. Similar to the previous point about the follow-up sample, this may mean our findings are not generalisable to the general population of smokers. Fourth, the survey during this study did not collect data on perceived stress levels; this would be a useful variable to consider in future research. Fifth, smokers may not have recalled (and reported) all quit attempts made in the 12 months preceding the follow-up survey [33, 34]. Sixth, the analyses pertaining to motivation to stop were cross-sectional and therefore the direction of the association is unclear.

## Implications for research, policy and practice

Future research to unpick the potential explanations for the positive association between smoking to relieve stress and high motivation to stop would benefit from deploying qualitative methods and/or Ecological Momentary Assessments in smokers' daily lives, allowing a closer examination of the temporal dynamics of competing (or synergistic) motives, evaluations and impulses [8, 35]. Further understanding of the relationship between smoking to relieve stress and future quit attempts is also required in studies with larger sample sizes. As many quit attempts are, on average, needed before quitting smoking successfully [36], a positive relationship between smoking to relieve stress and making a quit attempt in the future would likely have a positive impact on public health.

The finding that smoking to relieve stress is a commonly endorsed motive in smokers from both lower and higher SEP groups indicates that public health interventions may benefit from tackling such misperceptions about smoking and stress relief, with evidence from a recent systematic review and meta-analysis of longitudinal studies indicating that smoking cessation (compared with continued smoking) is associated with reduced stress, anxiety and depression [6]. Although the finding that smoking to relieve stress is associated with high motivation to stop (and possibly the likelihood of making a quit attempt in the future) may be perceived as a positive finding that does not warrant intervention, it may be more ethically defensible to highlight to smokers the large body of work indicating that smoking cessation leads to positive mental health outcomes and use this as a motive for stopping, thus correcting common misperceptions. In addition, if smoking is normatively perceived as a stress reducer, then it may encourage people to take up smoking. This merits further investigation.

At the same time (although not measured in the present study), research shows that many mental health professionals believe that smoking helps to alleviate stress and therefore refrain from offering appropriate smoking cessation support to their clients [37]. This suggests that

the tackling of common misperceptions also needs to occur among practitioners. Public health campaigns have often focused on the well-documented physical health aspects of smoking, with less attention paid to the mental health benefits of quitting, including reduced stress, anxiety and depression [6] but also increased levels of happiness [38, 39]. With increasing evidence for how to best counter misperceptions without inadvertently reinforcing them in the areas of vaccine hesitancy and climate change (e.g. normative feedback, framing messages in terms of gain) [40–42], interventions targeting healthcare professionals may benefit from drawing on this growing literature.

Finally, prior research shows that although smokers from lower SEP groups try to quit at a similar rate to smokers from higher SEP groups, they are less likely to succeed [43]. It is plausible that people from lower SEP groups endorse a confluence of motivational, cognitive and social factors for continued smoking (e.g. smoking to relieve stress and boredom, lower quitting self-efficacy, social aspects of smoking), which may interact to make it more difficult to quit [44]. Future research should examine the potential clustering of such factors and whether certain patterns are more pronounced in lower (compared with higher) SEP groups, as this is needed to help reduce smoking-related health inequalities.

## Conclusion

Smoking to relieve stress is a commonly endorsed motive and is associated with high motivation to stop but not significantly with the odds of making a future quit attempt, although the magnitude and direction of the effect was similar for both outcomes. There was no clear evidence of moderation by SEP, although data were insensitive to distinguish the alternative from the null hypothesis.

## Supporting information

**S1 Table. Unadjusted and adjusted odds ratios for the associations of smoking to relieve stress with i) high motivation to stop and ii) quit attempts in the next 12 months in the planned sensitivity analysis with housing tenure.**
(DOCX)

**S2 Table. Unadjusted and adjusted odds ratios for the associations of smoking to relieve stress with i) high motivation to stop and ii) quit attempts in the next 12 months in the unplanned sensitivity analysis with cigarettes per day coded as a categorical variable.**
(DOCX)

**S3 Table. Unadjusted and adjusted odds ratios for the associations of smoking to relieve stress with i) high motivation to stop and ii) quit attempts in the next 12 months in the higher SEP group.**
(DOCX)

**S4 Table. Unadjusted and adjusted odds ratios for the associations of smoking to relieve stress with i) high motivation to stop and ii) quit attempts in the next 12 months in the lower SEP group.**
(DOCX)

## Acknowledgments

We gratefully acknowledge the funding listed.

## Author Contributions

**Conceptualization:** Olga Perski, Maria Theodoraki, Sharon Cox, Loren Kock, Lion Shahab, Jamie Brown.

**Data curation:** Olga Perski.

**Formal analysis:** Olga Perski.

**Funding acquisition:** Lion Shahab, Jamie Brown.

**Investigation:** Olga Perski.

**Methodology:** Olga Perski.

**Project administration:** Olga Perski.

**Writing – original draft:** Olga Perski.

**Writing – review & editing:** Olga Perski, Maria Theodoraki, Sharon Cox, Loren Kock, Lion Shahab, Jamie Brown.

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
