## [Decision Letter · Decision Letter 0]

12 Jan 2022

PONE-D-21-18453Associations between smoking to relieve stress, motivation to stop and future quit attempts across the social spectrum: A population survey in EnglandPLOS ONE

Dear Dr. Perski,

Thank you for submitting your manuscript to PLOS ONE. After careful consideration, we feel that it has merit but does not fully meet PLOS ONE’s publication criteria as it currently stands. Therefore, we invite you to submit a revised version of the manuscript that addresses the points raised during the review process.

We look forward to receiving your revised manuscript.

Kind regards,

Jesse T. Kaye, PhD

Academic Editor

PLOS ONE

Journal Requirements:

3. Please include additional information regarding the survey or questionnaire used in the study and ensure that you have provided sufficient details that others could replicate the analyses. For instance, if you developed a questionnaire as part of this study and it is not under a copyright more restrictive than CC-BY, please include a copy, in both the original language and English, as Supporting Information. If the original language is written in non-Latin characters, for example Amharic, Chinese, or Korean, please use a file format that ensures these characters are visible.

4. Please state whether you validated the questionnaire prior to testing on study participants. Please provide details regarding the validation group within the methods section.

6. Thank you for stating the following in the Competing Interests section: 

"OP, MT, and LK have no conflicts of interest to declare. SC has provided expert consultancy to providers of UK life insurance and the pharmaceutical industry on matters relating to smoking cessations aids. LS has received a research grant and honoraria for a talk and travel expenses from manufacturers of smoking cessation medications (Pfizer and Johnson & Johnson). JB has received unrestricted research funding from Pfizer to study smoking cessation."

We note that you received funding from a commercial source: 'Pfizer and Johnson & Johnson'

7. Please note that in order to use the direct billing option the corresponding author must be affiliated with the chosen institute. Please either amend your manuscript to change the affiliation or corresponding author, or email us at plosone@plos.org with a request to remove this option.

8. PLOS requires an ORCID iD for the corresponding author in Editorial Manager on papers submitted after December 6th, 2016. Please ensure that you have an ORCID iD and that it is validated in Editorial Manager. To do this, go to ‘Update my Information’ (in the upper left-hand corner of the main menu), and click on the Fetch/Validate link next to the ORCID field. This will take you to the ORCID site and allow you to create a new iD or authenticate a pre-existing iD in Editorial Manager. Please see the following video for instructions on linking an ORCID iD to your Editorial Manager account: https://www.youtube.com/watch?v=_xcclfuvtxQ

Additional Editor Comments (if provided):

The authors report results of an interesting observational study examining the relations between self-reported stress-relief smoking motives and motivation to stop smoking and follow-up quit attempts. In particular, the authors are interested in whether SEP moderates these associations. These represent important scientific questions to inform movement toward achieving health equity in smoking treatment and intervention. The authors provide a detailed pre-registration on OSF in line with open science principles. Two expert reviewers provided their thoughtful critiques and recommendations below. I would encourage the authors to address their comments in detail, which I would consider a major revision.

The primary project seems well-designed, although additional methods details requested by both reviewers would enhance the manuscript and aid in interpretation of the data. One significant concern is the very small sample size at follow up and in particular that this sample may not be representative of the larger initial sample (e.g., especially on the SEP moderator variable of interest). As reviewer 1 notes, the limitations should not simply be listed but rather discuss how they constrain the interpretation of the results. For the small follow up sample size and representativeness, this should especially be discussed in greater detail and provide more cautious interpretation of this follow up data given these concerns (see reviewer 2). Furthermore, I would encourage the authors to be careful with their language in the discussion to how they interpret non-significant effects per reviewer 1’s recommendations. Additionally, the description of the results in the discussion should mirror the findings in the results section – it is not appropriate to imply group differences when the data analyses do not reveal significant differences (e.g., discussion first paragraph, last sentence).

Reviewers' comments:

Reviewer's Responses to Questions

**Comments to the Author**

1. Is the manuscript technically sound, and do the data support the conclusions?

Reviewer #1: Partly

Reviewer #2: Yes

2. Has the statistical analysis been performed appropriately and rigorously? 

Reviewer #1: Yes

Reviewer #2: Yes

3. Have the authors made all data underlying the findings in their manuscript fully available?

Reviewer #1: Yes

Reviewer #2: Yes

4. Is the manuscript presented in an intelligible fashion and written in standard English?

Reviewer #1: Yes

Reviewer #2: Yes

5. Review Comments to the Author

Reviewer #1: This is a well-designed study with interesting results for their well-justified research question. However, the paper would be greatly improved by the authors paying more attention to the interpretation of their results in the Discussion.

1. L33-42. Abstract. Revise the Results and Conclusion after addressing the suggestions below.

2. The introduction is clear and sets the seen well and justifies the research question well.

3. L188-190. Methods. It appears CPD has been treated as an interval variable rather than as a categorical variable, as for quit attempts in past year. This makes assumptions about the relationship between CPD and the outcomes which may not be justified, and a categorical variable may be more appropriate (and indeed they could consider using the combination measure of dependence Heaviness of Smoking index based on CPD and TTFC).

4. L198. It appears just the baseline was weighted to the population, not the cohort that was followed up. Please clarify, and explain why the cohort was not weighted if it was not.

5. L211-214. The description of the alternative SEP variable based on housing tenure would be better moved to paragraph describing covariates (ending L190).

6. L231-234. Results. This sentence only refers to p-values, not effect size or interval estimates. The authors should review how they report their results, especially results that are ‘not statistically significant’ in light of recommendations such as these and many more: Wasserstein RL, Lazar NA. The ASA statement on p‐values: context, process, and purpose. Am Stat 2016; 70: 129–133. Greenland S, McShane B. Scientists rise up against statistical significance. Nature 2019; 567: 305–307.

7. L244-248. The OR of the interaction term is not informative to readers. The authors could consider reporting separately the ORs for high and low SEP subgroups, perhaps in a supplementary table, to demonstrate the magnitude and direction of the effect modification (even though it is not statistically significant, and the Bayesian analysis suggested it was insensitive to detect an interaction effect).

8. L249-255. This paragraph would more logically sit before the paragraph above about interaction.

9. L251-255. Be more precise: start the sentence with ‘The adjusted association between the exposure and both outcome variables was unchanged…’ not ‘Results were materially …’. Drop the next two sentences as not germane, they are just about the adjusted association between an alternative covariate and the outcome.

10. Table 2. Consider just reporting an overarching p-value for the age and quit attempt categorical variables, rather than p-values for each category. Should also include the n for each analysis, and a footnote explaining what is adjusted for in the adjusted ORs – all variables. Consider also reporting the unadjusted association of the co-variates with the outcomes. Please check the journal guidelines if the precision of the reported p-values is correct: typically I would report 0.08 (not 0.084), and only go to 3 decimal points if less than 0.01.

11. Supplementary table – as above.

12. Discussion. Principal findings. Second and third sentence. Reconsider how you report non-significant findings. This does not mean that there is no association, just that you were unable to detect one. The point estimate of the association with making a quit attempt is of similar magnitude and direction as the association with motivation – however this is based on a smaller sample with less power. So don’t assume no impact, indeed consider discussing WHY there may also be an impact on quit attempts. The last sentence of this paragraph is inconsistent with the results – there is small non-significant difference between the groups: they are similar.

13. Discussion. Principal findings. Second paragraph. There is only scant discussion of possible reasons for the authors’ principal findings. The smoker’s belief that they are addicted is true, but the belief that they are motivated to stay smoking because they believe smoking helps them cope with stress is based on a misperception. The authors get close to explaining the link in the fourth sentence, but this needs to be stepped out more clearly. This should come before the previous sentence as that is about a methodological reason why this association may be spurious due to reverse causation. Might there also be a common factor that causes both the explanatory and outcome factor? The authors need to also provide some explanation for other principal findings, e.g. no difference in the belief that they are motivated to stay smoking because they believe smoking helps them cope with stress in the two SEP groups, before jumping to the need for more research, strengths and limitations, and implications for practice. These explanations need to better contextualise their new results with previous research.

14. Discussion. Strengths and limitations. Rather than just list these, the authors need to consider how they are likely to have altered their results and interpretation and implications of the results.

15. Discussion. Implications. First paragraph, second sentence. But if the perception is associated with increased motivation (and possibly increased likelihood of making a quit attempt in the future), this may not be a problem that needs to be corrected! Please discuss. In contrast, in the next sentence, this same misperception being held by health professionals definitely is a problem (which the authors have not measured or examined) and needs to be countered as it leads to less support. There is now increasing research evidence about how best counter such misperceptions without inadvertently reinforcing them, eg about vaccines and climate change, that could be referred to here.

16. Discussion. Implications. Second paragraph. The findings suggest that addressing misperceptions will be equity neutral, as similar association in high and low SEP. The paper does NOT address the association between stress and quitting, there are no implications from the results about addressing stress itself.

17. Discussion. Implications. Last paragraph. The first sentence is well substantiated, but the second sentence is not substantiated. Indeed, from other research, motivation seems mainly associated with initiating but not sustaining quit attempts.

18. Conclusion. The second sentence is correct and is a more correct description of the last clause of the first sentence.

Note that I am insufficiently familiar with Bayesian statistics to adequately review those methods.

Reviewer #2: The authors have presented an interesting manuscript that explored the relationship between using smoking to relieve stress and motivation to quit and future quit attempts in a UK sample of people 16 years and older. I think the authors did a good job laying out the premise for the analysis and I do think that there is merit in understanding how stress and coping with stress impacts smoking behaviors. The relationship is particularly salient among low SES populations as often we are inclined to advise people to quit but we don’t offer strategies to help them with the many stressors that lead them to smoke in the first place. I did have some questions about the methods and conclusions, which I will highlight below.

Methods

- Providing additional details on the Smoking Toolkit Study or a reference would be helpful. For example, I had the following questions:

o The sample was hybrid of probability and quota sample. Could you share how much was a probability based sample?

o Were respondents to the survey incentivized?

o How many waves are there for this survey?

o How were attempts made to reconnect with survey participants at 1 year follow-up? Why were there only 153 people at follow-up from such a large sample at baseline. Was the survey designed to be longitudinal? What steps were taken to ensure a higher follow-up rate?

- Quit attempt definition: We normally use the question, did you make a quit attempt lasting 24 hours or more as an indicator of a serious quit attempt. Has your question been validated? I also think future quit attempts is a bit confusing. These are smokers at baseline who made quit attempts during the study time period and reported them at 1 year follow-up. I might just say quit attempts at 1 year follow-up

- It seem like all of the smokers who were followed-up were still smoking at 1 year – were there any smokers who were abstinent?

- I appreciated that the authors presented the analysis using the Bayes factor and that does put the non-significant findings in perspective, however I do think the authors can clarify this section in the methods. For example, you could put the parameters for the BF for null hypothesis being true (BF close to 0), alternative hypothesis possible (BF close >1), and data insensitive (BF close to 1). This will allow for more clear interpretation of data in the results.

Results and conclusions

- I understand that the bayes factor for the interaction analysis showed that the data were insensitive to detect an association by SEP. I think it would help the reader to describe in more detail what the implications are of BF results that the data are insensitive to the association. From my understanding, it essentially means that a negative finding in your study does not rule out the possibility of an association in another study with a larger sample size and that this question merits consideration in larger studies. Perhaps including this in the first paragraph of discussion could be helpful.

- My main concern is around the follow-up sample being quite different from the overall sample at baseline. Therefore the conclusion of smoking for stress and quit attempt may only be relevant to a sub-sample that were followed up: older smoker, men, and those in higher SES groups. This does bring in some limitations to the conclusions – and not only were the results statistically non-significant but if they were significant then the might only apply to a sub-sample of smokers who were present at follow-up. I might make this more clear in the main findings of the study.

- Lastly, these were smokers who attempted to quit but were not successful so it does bring up the question about quit attempts being the mediator of successful quitting among those who smoke for stress. I.e., if we can demonstrate a connection that people who smoke for stress do attempt to quit and make multiple quit attempts, then that might be a marker of future successful quitting. Perhaps that point in the discussion could help frame these findings and provide a motivation for continuing to explore this association.

6. PLOS authors have the option to publish the peer review history of their article (what does this mean?). If published, this will include your full peer review and any attached files.

Reviewer #1: **Yes: **David Thomas

Reviewer #2: No

---

## [Author Response · Author response to Decision Letter 0]

25 Feb 2022

The manuscript now aligns with PLOS ONE’s style requirements.

We have now added the following to the “Ethics statement” on p.5 and the “Ethics approval and consent to participate” statement on p. 14:

“Respondents provided verbal informed consent prior to responding to the survey.”

3. Please include additional information regarding the survey or questionnaire used in the study and ensure that you have provided sufficient details that others could replicate the analyses. For instance, if you developed a questionnaire as part of this study and it is not under a copyright more restrictive than CC-BY, please include a copy, in both the original language and English, as Supporting Information. If the original language is written in non-Latin characters, for example Amharic, Chinese, or Korean, please use a file format that ensures these characters are visible.

We have included in the main body of the manuscript the exact wording of the survey items and response options used in the study.

4. Please state whether you validated the questionnaire prior to testing on study participants. Please provide details regarding the validation group within the methods section.

We have included details regarding the validation of the Motivation To Stop Scale on p.6-7. The remaining survey items are standard questions that have not all undergone formal validation but have been used in used in numerous papers that have been able to detect variations in quit attempt rates associated with mass media campaigns, and other population-level influences, as well as being the basis for the real-world evaluation of treatments, which in most cases replicated the results from RCTs.

https://pubmed.ncbi.nlm.nih.gov/31418449/

https://pubmed.ncbi.nlm.nih.gov/24322004/

https://academic.oup.com/ntr/article/22/9/1453/5524245?login=true

https://pubmed.ncbi.nlm.nih.gov/24372901/

5. We note that the grant information you provided in the ‘Funding Information’ and ‘Financial Disclosure’ sections do not match. When you resubmit, please ensure that you provide the correct grant numbers for the awards you received for your study in the ‘Funding Information’ section.

We have now added a ‘Funding information’ section to the manuscript, which matches the ‘Financial disclosure’ section.

6. Thank you for stating the following in the Competing Interests section: 

"OP, MT, and LK have no conflicts of interest to declare. SC has provided expert consultancy to providers of UK life insurance and the pharmaceutical industry on matters relating to smoking cessations aids. LS has received a research grant and honoraria for a talk and travel expenses from manufacturers of smoking cessation medications (Pfizer and Johnson & Johnson). JB has received unrestricted research funding from Pfizer to study smoking cessation." We note that you received funding from a commercial source: 'Pfizer and Johnson & Johnson.' Please provide an amended Competing Interests Statement that explicitly states this commercial funder, along with any other relevant declarations relating to employment, consultancy, patents, products in development, marketed products, etc. Within this Competing Interests Statement, please confirm that this does not alter your adherence to all PLOS ONE policies on sharing data and materials by including the following statement: "This does not alter our adherence to PLOS ONE policies on sharing data and materials.” (as detailed online in our guide for authors http://journals.plos.org/plosone/s/competing-interests). If there are restrictions on sharing of data and/or materials, please state these. Please note that we cannot proceed with consideration of your article until this information has been declared. Please include your amended Competing Interests Statement within your cover letter. We will change the online submission form on your behalf.

This has now been amended and added to the cover letter:

“OP, MT, SC and LK have no conflicts of interest to declare. LS has received a research grant and honoraria for a talk and travel expenses from manufacturers of smoking cessation medications (Pfizer and Johnson & Johnson). JB has received unrestricted research funding from Pfizer to study smoking cessation. This does not alter our adherence to PLOS ONE’s policies on sharing data and materials.”

7. Please note that in order to use the direct billing option the corresponding author must be affiliated with the chosen institute. Please either amend your manuscript to change the affiliation or corresponding author, or email us at plosone@plos.org with a request to remove this option.

I’ve asked the PLOS ONE team to merge multiple accounts (required to amend this information).

8. PLOS requires an ORCID iD for the corresponding author in Editorial Manager on papers submitted after December 6th, 2016. Please ensure that you have an ORCID iD and that it is validated in Editorial Manager. To do this, go to ‘Update my Information’ (in the upper left-hand corner of the main menu), and click on the Fetch/Validate link next to the ORCID field. This will take you to the ORCID site and allow you to create a new iD or authenticate a pre-existing iD in Editorial Manager. Please see the following video for instructions on linking an ORCID iD to your Editorial Manager account: https://www.youtube.com/watch?v=_xcclfuvtxQ

I’ve asked the PLOS ONE team to merge multiple accounts (required to amend this information).

This has now been amended.

Additional Editor Comments (if provided):

The authors report results of an interesting observational study examining the relations between self-reported stress-relief smoking motives and motivation to stop smoking and follow-up quit attempts. In particular, the authors are interested in whether SEP moderates these associations. These represent important scientific questions to inform movement toward achieving health equity in smoking treatment and intervention. The authors provide a detailed pre-registration on OSF in line with open science principles. Two expert reviewers provided their thoughtful critiques and recommendations below. I would encourage the authors to address their comments in detail, which I would consider a major revision.

We thank the Editor and the two reviewers for their helpful feedback on our manuscript.

The primary project seems well-designed, although additional methods details requested by both reviewers would enhance the manuscript and aid in interpretation of the data. 

We have now added methodological details about the survey items, sampling strategy and incentivisation in the Methods section.

One significant concern is the very small sample size at follow up and in particular that this sample may not be representative of the larger initial sample (e.g., especially on the SEP moderator variable of interest). As reviewer 1 notes, the limitations should not simply be listed but rather discuss how they constrain the interpretation of the results. For the small follow up sample size and representativeness, this should especially be discussed in greater detail and provide more cautious interpretation of this follow up data given these concerns (see reviewer 2). 

We provide a more elaborate discussion as to how the small sample size and lack of representativeness constrain the interpretation of the results on p.14.

Furthermore, I would encourage the authors to be careful with their language in the discussion to how they interpret non-significant effects per reviewer 1’s recommendations. Additionally, the description of the results in the discussion should mirror the findings in the results section – it is not appropriate to imply group differences when the data analyses do not reveal significant differences (e.g., discussion first paragraph, last sentence).

We have now modified the framing of the non-significant results throughout as per reviewer 1’s suggestion and have removed the sentence alluding to group differences in the Discussion on p.12.

Reviewer 1

This is a well-designed study with interesting results for their well-justified research question. However, the paper would be greatly improved by the authors paying more attention to the interpretation of their results in the Discussion.

We thank the reviewer for their constructive feedback on our manuscript, which we believe has strengthened it.

1. L33-42. Abstract. Revise the Results and Conclusion after addressing the suggestions below.

This has now been amended:

“Smoking to relieve stress is a commonly endorsed motive and is associated with high motivation to stop but not significantly with the odds of making a future quit attempt, although the magnitude and direction of the effect was similar for both outcomes. There was no clear evidence of moderation by SEP, although data were insensitive to distinguish the alternative from the null hypothesis.”

2. The introduction is clear and sets the seen well and justifies the research question well.

Thank you.

3. L188-190. Methods. It appears CPD has been treated as an interval variable rather than as a categorical variable, as for quit attempts in past year. This makes assumptions about the relationship between CPD and the outcomes which may not be justified, and a categorical variable may be more appropriate (and indeed they could consider using the combination measure of dependence Heaviness of Smoking index based on CPD and TTFC).

As suggested, we have conducted sensitivity analyses with CPD entered as a categorical variable, with associations remaining robust (p.10):

“In unplanned sensitivity analyses with CPD coded as a categorical variable, the associations were similarly unchanged (see S1 Table 2).”

4. L198. It appears just the baseline was weighted to the population, not the cohort that was followed up. Please clarify, and explain why the cohort was not weighted if it was not.

As is common within the STS, we did not observe important differences between the unweighted and weighted baseline samples (see Table 1). Therefore, the inferential analyses remain based on unweighted data, as planned. We also provide details on the unweighted and weighted follow-up sample in Table 1 (with differences between these being similarly small).

5. L211-214. The description of the alternative SEP variable based on housing tenure would be better moved to paragraph describing covariates (ending L190).

This has now been moved, as suggested.

6. L231-234. Results. This sentence only refers to p-values, not effect size or interval estimates. The authors should review how they report their results, especially results that are ‘not statistically significant’ in light of recommendations such as these and many more: Wasserstein RL, Lazar NA. The ASA statement on p‐values: context, process, and purpose. Am Stat 2016; 70: 129–133. Greenland S, McShane B. Scientists rise up against statistical significance. Nature 2019; 567: 305–307.

We agree with this (which is – we hope – at least partly reflected by us opting to complement the frequentist analyses with the calculation of Bayes Factors) and have reframed the reporting of the non-significant results throughout – e.g., in the Abstract:

“Smoking to relieve stress is a commonly endorsed motive and is associated with high motivation to stop but not significantly with the odds of making a future quit attempt, although the magnitude and direction of the effect was similar for both outcomes. There was no clear evidence of moderation by SEP, although data were insensitive to distinguish the alternative from the null hypothesis.”

7. L244-248. The OR of the interaction term is not informative to readers. The authors could consider reporting separately the ORs for high and low SEP subgroups, perhaps in a supplementary table, to demonstrate the magnitude and direction of the effect modification (even though it is not statistically significant, and the Bayesian analysis suggested it was insensitive to detect an interaction effect).

We now report the ORs for the higher and lower SEP groups separately in S1 Tables 3a and 3b (referred to on p.10).

8. L249-255. This paragraph would more logically sit before the paragraph above about interaction.

This has now been amended.

9. L251-255. Be more precise: start the sentence with ‘The adjusted association between the exposure and both outcome variables was unchanged…’ not ‘Results were materially …’. Drop the next two sentences as not germane, they are just about the adjusted association between an alternative covariate and the outcome.

This has now been amended.

10. Table 2. Consider just reporting an overarching p-value for the age and quit attempt categorical variables, rather than p-values for each category. 

Thank you – as the relationships appear non-linear, we prefer to present the ORs and p-values for each category.

Should also include the n for each analysis, and a footnote explaining what is adjusted for in the adjusted ORs – all variables. 

We now report the N for each analysis and include a footnote which indicates the covariates adjusted for.

Consider also reporting the unadjusted association of the co-variates with the outcomes. 

We now report the unadjusted associations of the covariates with the outcomes in Table 1.

Please check the journal guidelines if the precision of the reported p-values is correct: typically I would report 0.08 (not 0.084), and only go to 3 decimal points if less than 0.01.

PLOS ONE’s guidelines only state that exact p-values (if greater than 0.001) should be reported but do not mention the number of decimal points.

11. Supplementary table – as above.

This has now been amended, as suggested.

12. Discussion. Principal findings. Second and third sentence. Reconsider how you report non-significant findings. This does not mean that there is no association, just that you were unable to detect one. The point estimate of the association with making a quit attempt is of similar magnitude and direction as the association with motivation – however this is based on a smaller sample with less power. So don’t assume no impact, indeed consider discussing WHY there may also be an impact on quit attempts. The last sentence of this paragraph is inconsistent with the results – there is small non-significant difference between the groups: they are similar.

We have now changed the wording pertaining to the non-significant results on p.12. We agree that the last sentence is inconsistent with the results and have removed it.

“Although non-significant, both the direction and magnitude of the association of smoking to relieve stress with future quit attempts were similar to those for the association with high motivation to stop. We did not observe significant moderation by SEP; however, the Bayes Factors indicated that data were insensitive to detect an association (as opposed to lending support to the null hypothesis of no association). For this to be known, other studies with larger samples need to be conducted to detect any potential moderation effects by SEP.”

13. Discussion. Principal findings. Second paragraph. There is only scant discussion of possible reasons for the authors’ principal findings. The smoker’s belief that they are addicted is true, but the belief that they are motivated to stay smoking because they believe smoking helps them cope with stress is based on a misperception. The authors get close to explaining the link in the fourth sentence, but this needs to be stepped out more clearly. This should come before the previous sentence as that is about a methodological reason why this association may be spurious due to reverse causation. Might there also be a common factor that causes both the explanatory and outcome factor? 

Thank you – we have now moved the methodological consideration about reverse causation to the end of the paragraph and included further discussion of possible reasons for the principal results.

“First, people who smoke to relieve stress may evaluate their smoking experience as biological or instrumental (as opposed to pleasurable), which may be perceived as unpleasant or annoying, hence leading to greater motivation to stop. Second, those who smoke to relieve stress may find smoking enjoyable but at the same time, find it a source of dissonance to use smoking as a ‘crutch’ to cope with stress, hence leading to greater motivation to stop. Third, it is also possible that there is a common factor (e.g., depression) that gives rise both to the belief that smoking helps to relieve stress and high motivation to stop. Previous research has found that smokers with more severe (compared with less severe) levels of depression report lower quitting self-efficacy (which may be influenced by positive outcome expectations, such as the belief that smoking helps to cope with stress) but higher motivation to stop [33].”

The authors need to also provide some explanation for other principal findings, e.g. no difference in the belief that they are motivated to stay smoking because they believe smoking helps them cope with stress in the two SEP groups, before jumping to the need for more research, strengths and limitations, and implications for practice. These explanations need to better contextualise their new results with previous research.

Thank you – we now discuss this on p.13-14:

“The finding that smokers from both higher and lower SEP groups endorse the belief that smoking helps to relieve stress may indicate that smokers are not only more/less likely to endorse this belief depending on their level of actual stressors (which may, on average, be higher in smokers from lower SEP groups). Instead, this may be related to people’s perceived ability to cope with stressors. Smokers across the social spectrum may perceive themselves to be better/worse equipped to cope with stress (irrespective of whether they actually are or not), thus influencing their beliefs about the role of smoking (e.g. positive outcome expectancies).”

14. Discussion. Strengths and limitations. Rather than just list these, the authors need to consider how they are likely to have altered their results and interpretation and implications of the results.

Thank you – we now elaborate on this in the Discussion on p.14:

“Second, the follow-up sample differed from the baseline sample on important characteristics, including SEP. This means that, had our findings pertaining to the follow-up sample been statistically significant, they may not be generalisable to the general population of smokers. In addition, if those who were more likely to be followed up (i.e. older smokers, men, those from higher SEP groups) were also more or less likely to endorse the belief that they smoke to relieve stress, this may have distorted any association with the outcome. Third, although the Smoking Toolkit Study is a nationally representative survey on key sociodemographic and smoking characteristics, it may not be representative with regards to psychological characteristics such as motivation to stop or the motive that smoking helps to relieve stress and it also may not reach smokers who are the most disadvantaged. Similar to the previous point about the follow-up sample, this may mean our findings are not generalisable to the general population of smokers.”

15. Discussion. Implications. First paragraph, second sentence. But if the perception is associated with increased motivation (and possibly increased likelihood of making a quit attempt in the future), this may not be a problem that needs to be corrected! Please discuss. 

We now discuss this on p.15:

“Although the finding that smoking to relieve stress is associated with high motivation to stop (and possibly the likelihood of making a quit attempt in the future) may be perceived as a positive finding that does not warrant intervention, it may be more ethically defensible to highlight to smokers the large body of work indicating that smoking cessation leads to positive mental health outcomes and use this as a motive for stopping, thus correcting common misperceptions. In addition, if smoking is normatively perceived as a stress reducer, then it may encourage people to take up smoking. This merits further investigation.”

In contrast, in the next sentence, this same misperception being held by health professionals definitely is a problem (which the authors have not measured or examined) and needs to be countered as it leads to less support. There is now increasing research evidence about how best counter such misperceptions without inadvertently reinforcing them, eg about vaccines and climate change, that could be referred to here.

We have added the following on p.15:

“At the same time (although not measured in the present study), research shows that many mental health professionals believe that smoking helps to alleviate stress and therefore refrain from offering appropriate smoking cessation support to their clients [38]. This suggests that the tackling of common misperceptions also needs to occur among practitioners. Public health campaigns have often focused on the well-documented physical health aspects of smoking, with less attention paid to the mental health benefits of quitting, including reduced stress, anxiety and depression [6] but also increased levels of happiness [39,40]. With increasing evidence for how to best counter misperceptions without inadvertently reinforcing them in the areas of vaccine hesitancy and climate change (e.g. normative feedback, framing messages in terms of gain) [41–43], interventions targeting healthcare professionals may benefit from drawing on this growing literature.”

16. Discussion. Implications. Second paragraph. The findings suggest that addressing misperceptions will be equity neutral, as similar association in high and low SEP. The paper does NOT address the association between stress and quitting, there are no implications from the results about addressing stress itself.

We agree that this paragraph goes beyond the current findings and have therefore removed it.

17. Discussion. Implications. Last paragraph. The first sentence is well substantiated, but the second sentence is not substantiated. Indeed, from other research, motivation seems mainly associated with initiating but not sustaining quit attempts.

Thank you – this has now been amended to reflect the evidence:

“It is plausible that people from lower SEP groups endorse a confluence of motivational, cognitive and social factors for continued smoking (e.g. smoking to relieve stress and boredom, lower quitting self-efficacy, social aspects of smoking), which may interact to make it more difficult to quit [45]. Future research should examine the potential clustering of such factors and whether certain patterns are more pronounced in lower (compared with higher) SEP groups, as this is needed to help reduce smoking-related health inequalities.”

18. Conclusion. The second sentence is correct and is a more correct description of the last clause of the first sentence.

We have now amended this, as follows:

“Smoking to relieve stress is a commonly endorsed motive and is associated with high motivation to stop but not significantly with the odds of making a future quit attempt, although the magnitude and direction of the effect was similar for both outcomes. There was no clear evidence of moderation by SEP, although data were insensitive to distinguish the alternative from the null hypothesis.”

Note that I am insufficiently familiar with Bayesian statistics to adequately review those methods.

Thank you.

Reviewer 2

The authors have presented an interesting manuscript that explored the relationship between using smoking to relieve stress and motivation to quit and future quit attempts in a UK sample of people 16 years and older. I think the authors did a good job laying out the premise for the analysis and I do think that there is merit in understanding how stress and coping with stress impacts smoking behaviors. The relationship is particularly salient among low SES populations as often we are inclined to advise people to quit but we don’t offer strategies to help them with the many stressors that lead them to smoke in the first place. I did have some questions about the methods and conclusions, which I will highlight below.

We thank the reviewer for their helpful feedback on our manuscript, which we believe has strengthened it.

Methods

- Providing additional details on the Smoking Toolkit Study or a reference would be helpful. For example, I had the following questions:

o The sample was hybrid of probability and quota sample. Could you share how much was a probability based sample?

We have now provided more information about the sampling strategy in the Methods on p.5, which clarifies why we cannot estimate how much was a probability-based sample:

“Interviews are held with one household member in selected geographic output areas until quotas are fulfilled. The quotas are based on factors influencing the probability of being at home (i.e. working status, age and gender). This hybrid form of random probability and quota sampling is considered superior to conventional quota sampling. Here, the choice of households to approach is limited by the random allocation of small output areas and rather than being sent to specific households in advance, interviewers can choose which households within these small geographic areas are most likely to fulfil their quotas. Therefore, unlike random probability sampling, it is not appropriate to record the response rate in the STS.”

o Were respondents to the survey incentivized?

We have now clarified this on p.5:

“Follow-up respondents were not remunerated.”

o How many waves are there for this survey?

We have now clarified the number of waves on p.5:

“Data included in the present study were collected across four waves (139-142) from respondents surveyed between April and July 2018 (when smoking to relieve stress was included in the STS).”

o How were attempts made to reconnect with survey participants at 1 year follow-up? Why were there only 153 people at follow-up from such a large sample at baseline. Was the survey designed to be longitudinal? What steps were taken to ensure a higher follow-up rate?

We have now clarified this on p.5:

“Follow-up respondents are not remunerated. Up to eight attempts are made to follow up each consenting respondent, with calls made at different times of day across weekdays and weekends.”

We have also clarified in the Discussion on p.13-14 that the small sample size for the second research question was at least partly due to limited funding:

“First, the sample size for the prospective association of smoking to relieve stress and future quit attempts was small. However, this was pre-determined by the period for which the funding was available (i.e., the question about smoking motives – including smoking to relieve stress – was included only in a few STS waves)…”

- Quit attempt definition: We normally use the question, did you make a quit attempt lasting 24 hours or more as an indicator of a serious quit attempt. Has your question been validated? 

Although this survey item has not been formally validated, the question was cognitively tested when establishing the survey in 2006, and has been used in numerous papers that have been able to detect variations in quit attempt rates associated with mass media campaigns, and other population-level influences, as well as being the basis for the real-world evaluation of treatments, which in most cases replicated the results from RCTs.

https://pubmed.ncbi.nlm.nih.gov/31418449/

https://pubmed.ncbi.nlm.nih.gov/24322004/

https://academic.oup.com/ntr/article/22/9/1453/5524245?login=true

https://pubmed.ncbi.nlm.nih.gov/24372901/

I also think future quit attempts is a bit confusing. These are smokers at baseline who made quit attempts during the study time period and reported them at 1 year follow-up. I might just say quit attempts at 1 year follow-up

This has been amended throughout.

- It seem like all of the smokers who were followed-up were still smoking at 1 year – were there any smokers who were abstinent?

We did not examine the smoking status of those followed up at 12 months as this was not one of our research questions.

- I appreciated that the authors presented the analysis using the Bayes factor and that does put the non-significant findings in perspective, however I do think the authors can clarify this section in the methods. For example, you could put the parameters for the BF for null hypothesis being true (BF close to 0), alternative hypothesis possible (BF close >1), and data insensitive (BF close to 1). This will allow for more clear interpretation of data in the results.

We have now clarified the interpretation of the Bayes Factors on p.8:

“Bayes Factors (BFs) can be interpreted as follows: BFs >3 lend support to the alternative hypothesis (over the null), BFs <1/3 lend support to the null, and BFs in-between reflect data being insensitive to distinguish the two.”

Results and conclusions

- I understand that the bayes factor for the interaction analysis showed that the data were insensitive to detect an association by SEP. I think it would help the reader to describe in more detail what the implications are of BF results that the data are insensitive to the association. From my understanding, it essentially means that a negative finding in your study does not rule out the possibility of an association in another study with a larger sample size and that this question merits consideration in larger studies. Perhaps including this in the first paragraph of discussion could be helpful.

Thank you for this helpful suggestion – we have now added this on p.13:

“We did not observe significant moderation by SEP; however, the Bayes Factors indicated that data were insensitive to detect an association (as opposed to lending support to the null hypothesis of no association). This means that it is plausible that another study with a larger sample may pick up moderation effects by SEP (if they exist).”

- My main concern is around the follow-up sample being quite different from the overall sample at baseline. Therefore the conclusion of smoking for stress and quit attempt may only be relevant to a sub-sample that were followed up: older smoker, men, and those in higher SES groups. This does bring in some limitations to the conclusions – and not only were the results statistically non-significant but if they were significant then the might only apply to a sub-sample of smokers who were present at follow-up. I might make this more clear in the main findings of the study.

Thank you – we now elaborate on this in the Discussion on p.14:

“Second, the follow-up sample differed from the baseline sample on important characteristics, including SEP. This means that, had our findings pertaining to the follow-up sample been statistically significant, they may not be generalisable to the general population of smokers. In addition, if those who were more likely to be followed up (i.e. older smokers, men, those from higher SEP groups) were also more or less likely to endorse the belief that they smoke to relieve stress, this may have distorted any association with the outcome.”

- Lastly, these were smokers who attempted to quit but were not successful so it does bring up the question about quit attempts being the mediator of successful quitting among those who smoke for stress. I.e., if we can demonstrate a connection that people who smoke for stress do attempt to quit and make multiple quit attempts, then that might be a marker of future successful quitting. Perhaps that point in the discussion could help frame these findings and provide a motivation for continuing to explore this association.

Thank you – we did not examine respondents’ smoking status at the 12-month follow-up (only if they had attempted to stop). However, we believe this is a useful point to highlight and have added the following in the Discussion on p.14: 

“As many quit attempts are, on average, needed before quitting smoking successfully [36], a positive relationship between smoking to relieve stress and making a quit attempt in the future would likely have a positive impact on public health.”

---

## [Decision Letter · Decision Letter 1]

2 May 2022

Associations between smoking to relieve stress, motivation to stop and quit attempts across the social spectrum: A population survey in England

PONE-D-21-18453R1

Dear Dr. Perski,

We’re pleased to inform you that your manuscript has been judged scientifically suitable for publication and will be formally accepted for publication once it meets all outstanding technical requirements.

Kind regards,

Jesse T. Kaye, PhD

Academic Editor

PLOS ONE

Additional Editor Comments (optional):

Thank you for resubmitting this revised manuscript for publication at PLOS ONE. The authors responded very thoroughly to the reviewers initial round of comments and the manuscript is improved as a result. Methods are explained in greater detail and the discussion of study findings is more comprehensive and also appropriately constrained by the data analysis and study limitations. This will be an important contribution to the literature. One of the original reviewers was able to review the revised manuscript and offered the suggestion to include the omnibus p-value for the categorical variables with multiple groups (age and number of quit attempts). The authors may add this to the table during copy editing if they would like (in addition, not instead of the individual p values). I agree it would enhance the table, but not absolutely required. Thank you for conducting this research to better understand the relations between stress coping motives and smoking cessation motivation.

Reviewers' comments:

Reviewer's Responses to Questions

**Comments to the Author**

1. If the authors have adequately addressed your comments raised in a previous round of review and you feel that this manuscript is now acceptable for publication, you may indicate that here to bypass the “Comments to the Author” section, enter your conflict of interest statement in the “Confidential to Editor” section, and submit your "Accept" recommendation.

Reviewer #1: (No Response)

2. Is the manuscript technically sound, and do the data support the conclusions?

Reviewer #1: Yes

3. Has the statistical analysis been performed appropriately and rigorously? 

Reviewer #1: Yes

4. Have the authors made all data underlying the findings in their manuscript fully available?

Reviewer #1: Yes

5. Is the manuscript presented in an intelligible fashion and written in standard English?

Reviewer #1: Yes

6. Review Comments to the Author

Reviewer #1: #10. Table 2. I agree relationships non-linear and a categorical variable based on series of dummy dummy variables is appropriate. I do not suggest a continuous variable. However most statistical programs (eg Stata) enable a post-estimation command which provides a p-value for the addition of the overall categorical variable. Not just the individual p-values comparing categories to the reference. I suggest this be included not the individual p-values.

7. PLOS authors have the option to publish the peer review history of their article (what does this mean?). If published, this will include your full peer review and any attached files.

Reviewer #1: No

---

## [Editor Report · Acceptance letter]

9 May 2022

PONE-D-21-18453R1 

Associations between smoking to relieve stress, motivation to stop and quit attempts across the social spectrum: A population survey in England 

Dear Dr. Perski:

I'm pleased to inform you that your manuscript has been deemed suitable for publication in PLOS ONE. Congratulations! Your manuscript is now with our production department. 

Kind regards, 

on behalf of

Dr. Jesse T. Kaye 

Academic Editor

PLOS ONE